# Novel Clinical *Campylobacter jejuni* Infection Models Based on Sensitization of Mice to Lipooligosaccharide, a Major Bacterial Factor Triggering Innate Immune Responses in Human Campylobacteriosis

**DOI:** 10.3390/microorganisms8040482

**Published:** 2020-03-28

**Authors:** Soraya Mousavi, Stefan Bereswill, Markus M. Heimesaat

**Affiliations:** Institute of Microbiology, Infectious Diseases and Immunology, Gastrointestinal Microbiology Research Group, Charité—University Medicine Berlin, Corporate Member of Freie Universität Berlin, Humboldt-Universität zu Berlin, and Berlin Institute of Health, 12203 Berlin, Germany; soraya.mousavi@charite.de (S.M.); stefan.bereswill@charite.de (S.B.)

**Keywords:** *Campylobacter jejuni*, campylobacteriosis, lipooligosaccharide (LOS), pathogenicity, intestinal pathogenesis, toll-like receptor

## Abstract

Human *Campylobacter jejuni* infections inducing campylobacteriosis including post-infectious sequelae such as Guillain-Barré syndrome and reactive arthritis are rising worldwide and progress into a global burden of high socioeconomic impact. Intestinal immunopathology underlying campylobacteriosis is a classical response of the innate immune system characterized by the accumulation of neutrophils and macrophages which cause tissue destruction, barrier defects and malabsorption leading to bloody diarrhea. Clinical studies revealed that enteritis and post-infectious morbidities of human *C. jejuni* infections are strongly dependent on the structure of pathogenic lipooligosaccharides (LOS) triggering the innate immune system via Toll-like-receptor (TLR)-4 signaling. Compared to humans, mice display an approximately 10,000 times weaker TLR-4 response and a pronounced colonization resistance (CR) against *C. jejuni* maintained by the murine gut microbiota. In consequence, investigations of campylobacteriosis have been hampered by the lack of experimental animal models. We here summarize recent progress made in the development of murine *C. jejuni* infection models that are based on the abolishment of CR by modulating the murine gut microbiota and by sensitization of mice to LOS. These advances support the major role of LOS driven innate immunity in pathogenesis of campylobacteriosis including post-infectious autoimmune diseases and promote the preclinical evaluation of novel pharmaceutical strategies for prophylaxis and treatment.

## 1. Introduction

The foodborne bacterial pathogen *Campylobacter jejuni* is recognized as one of the leading causes of infectious bacterial enteric infections worldwide [1,2,3,4,5,6,7]. Since 2005, campylobacteriosis has been the most frequently reported bacterial zoonosis in the European Union, exceeding salmonellosis by a continuously increasing number of cases [8]. Among more than 30 *Campylobacter* species and subspecies, *C. jejuni* and *C. coli* are the most common subpopulations causing human infections [9].

The Gram-negative, slender, spirally-curved, flagellated *C. jejuni* bacteria are highly motile and grow under microaerophilic conditions in a temperature range between 37 and 42 °C [10,11,12]. *C. jejuni* is present in surface waters and forms part of the natural intestinal microbiota of a wide range of wild animals as well as of agriculturally essential mammals and birds, especially poultry [9,11]. Recently, a study on pathogen isolates from humans and chicken confirmed the transmission of *C. jejuni* between the two species, underlining the significance of poultry as a source of human *C. jejuni* infections [13]. In fact, in the majority of disease cases, humans become infected via the consumption of undercooked meat of contaminated livestock animals or by ingestion of raw milk and surface water containing *C. jejuni* [14,15].

Depending on the *Campylobacter* strain and the host immune status, patients might present with a highly acute and severe symptom complex varying from watery diarrhea without fever and/or abdominal cramps to severe campylobacteriosis characterized by purulent bloody inflammatory diarrhea and systemic inflammatory responses including fever [9,16,17,18]. The infection is usually self-limiting and lasts for several days to two weeks [19,20]. However, in the minority of cases, post-infectious sequelae such as Guillain-Barré syndrome (GBS), Miller Fisher syndrome (MFS), reactive arthritis (RA) or chronic intestinal inflammatory morbidities including inflammatory bowel disease (IBD), irritable bowel syndrome (IBS) or celiac disease might develop [4,9,18,21,22,23]. A clinical study revealed that both the severity of campylobacteriosis and the development of post-infectious sequelae are significantly associated with sialylated lipooligosaccharide (LOS) structures localized in the outer cell membrane of *C. jejuni* [22]. Although the O-antigen characteristic of bacterial lipopolysaccharide (LPS) is missing in *C. jejuni* LOS [24,25], the structural variability of LOS provides the basis for the highly variable disease manifestation in humans. This finding was of great importance for the molecular understanding of the substantial role of *C. jejuni* LOS in intestinal immunopathogenesis of campylobacteriosis, which will be discussed further. 

## 2. Basic Concept and Aim of This Review Article

Similar to the vast majority of bacterial enteric pathogens causing inflammatory diseases in the gastrointestinal tract, *C. jejuni* enters the gut via ingestion of food contaminated with a low number of live bacteria [26]. After replication at body temperature and establishment of a primary population, the highly motile *C. jejuni* pass the barriers of the viscous mucus layer (Figure 1) and the epithelial cell lining with the help of polar flagella, adhesins, and invasins including potent proteases such as HtrA further supporting transcellular migration of the bacteria [27,28,29,30]. Most recently, the type VI secretion system (T6SS) of *C. jejuni* was discovered as a factor which might be involved in virulence. This system enables the contact-dependent secretion of effector proteins into host cells and even other bacteria [31]. However, the role of T6SS in campylobacteriosis is still unclear. Whereas results from several studies suggested that T6SS is associated with more severe disease [31,32], a recent clinical study demonstrated that the T6SS does not contribute to the severity of campylobacteriosis, as shown by analysis of human patients infected with T6SS negative and positive *C. jejuni* strains [33]. In contrast to the following inflammatory response, these “barrier breaking bacterial factors” are very well investigated at the molecular level and it has been established for decades that motility, adhesion and invasion are essential for *C. jejuni* pathogenicity and virulence (Figure 1). However, the analysis of biopsies taken from human patients revealed that *C. jejuni* reaching the lamina propria and the sub-epithelial tissues initiate a pronounced innate immune response, characterized by massive conglomerates of macrophages and neutrophilic granulocytes leading to acute inflammation [4,34,35,36,37]. The production of toxic oxygen radicals and cytokines over the course of this innate immune response further leads to apoptosis, tissue destruction and ulcerations, which finally pave the way for sodium malabsorption, followed by water efflux and bloody diarrhea [37] (Figure 1). 

Given the absence of potent *C. jejuni* toxins common to all infecting bacterial strains, the full-blown symptom complex of campylobacteriosis is caused by the innate immune responses induced by sialylated LOS variants A, B, and C, as shown in a clinical study focusing on factors mediating human campylobacteriosis [23]. In this aspect, campylobacteriosis mirrors key features of the immunopathogenesis during meningitis and urethritis caused by pathogenic *Neisseria meningitidis* and *N. gonorrhoeae*, respectively, both of which express LOS as a major pathogenicity factor on their bacterial surface [38] (Figure 1).

In addition to its functions in the initiation and propagation of intestinal inflammation the LOS of *C. jejuni* (Figure 2) has been identified as an essential virulence factor playing pivotal roles in host cell adhesion and invasion in vitro, as well as in evasion of host defense reactions [11,24,39,40,41]. The important role of LOS was most recently confirmed in a clinical study investigating the intestinal immunopathology in *C. jejuni* infected humans, which revealed that bacterial LOS is the master regulator of the innate immune responses in the onset and progress of human campylobacteriosis [37]. Structurally, LOS participates in the management of bacterial outer membrane stability and protects the *C. jejuni* cells from environmental stress conditions including host immune responses [24,42]. However, further analysis of LOS functions in campylobacteriosis has been hampered by the lack of suitable murine infection models. 

In contrast to humans, conventional mice are well protected from *C. jejuni* infection and do not develop clinical signs after experimental intestinal colonization [43]. This can be attributed to the fact that mice are approximately 10,000-fold less responsive to Toll-like receptor (TLR)-4 ligands, mainly LOS and LPS [44] (Figure 3). Furthermore, the gastrointestinal microbiota of conventional mice mediates a strong colonization resistance (CR) to *C. jejuni* [45,46,47]. In line with other investigators, we have recently shown that conventional mice bred in our specific pathogen-free (SPF) facilities are protected from stable gastrointestinal *C. jejuni* colonization even upon peroral infection with high doses. Upon modifying the murine gut microbiota (i.e., its virtual depletion) following broad-spectrum antibiotic treatment, and also upon re-association with human as opposed to murine gut microbiota by fecal microbiota transplantation, these mice can be effectively colonized by the pathogen upon peroral challenge and display typical histopathologic pro-inflammatory features of campylobacteriosis in their intestines [46,47,48,49], whereas the classic symptoms such as abdominal cramps, watery, or bloody diarrhea seen in infected humans were missing in conventional wildtype mice without genetic manipulations [46] (Figure 4).

In order to generate murine infection models mirroring clinical features of severe campylobacteriosis, researchers including our group—independently from each other—modified the microbiota in order to overcome CR, as well as LOS sensitivity via different genetic manipulations or by zinc depletion, which all in turn affect LOS/TLR-4 dependent signaling pathways [50,51,52,53,54] (Figure 4). 

Although experimental advances have been achieved for a better understanding of human campylobacteriosis, it is not yet fully understood how the host response itself is established at the cellular and molecular levels. However, several studies have suggested that the structure of *C. jejuni* LOS is indeed responsible for the outcome of infection and for the development of post-infectious sequelae [22,55]. In this review, we aim to summarize the development of novel clinical murine models of *C. jejuni* infection and their successful application in the study of major roles of *C. jejuni* LOS in the immunopathogenesis of campylobacteriosis (summarized in Table 1 and Table 2). 

## 3. Human Campylobacteriosis

Human *C. jejuni* infections are mostly self-limiting [9] and resolve within approximately 7 to 14 days [19,35], whereas some patients develop a more chronic form of *Campylobacter* colitis, however [65]. Interestingly, recent studies revealed that *C. jejuni* infection might increase the risk for development and relapses of IBD, IBS or celiac disease [66].

The development of *C. jejuni*-induced human enterocolitis occurs at different stages characterized by distinct immune responses upon infection [35]. In the early stage (i.e., the first week, during acute enterocolitis), epithelial and pathological changes predominantly in distal parts of the large intestinal tract are caused by a massive accumulation of macrophages and neutrophilic granulocytes in crypts and in the lamina propria (Figure 1). In the later stage (i.e., the second week), the symptoms of inflammation become less pronounced and the regeneration of the affected epithelia starts. After two weeks, in the residual stage, the inflammatory changes are almost completely resolved. Interestingly, regardless of the infection stage, *C. jejuni* bacteria have been detected immunohistochemically in colonic biopsies derived from patients with *C. jejuni* positive stool cultures [35]. 

Despite the described clinical picture, it is still neither completely clear at the cellular nor the molecular level how *C. jejuni* induces the macrophages and neutrophilic granulocytes driven inflammatory responses in humans. A multitude of in vitro studies confirmed that, in addition to the above-mentioned pathogenicity factors, various virulence factors including invasive properties, oxidative stress defense, cytolethal-distending toxin production, and iron acquisition, are required for full *C. jejuni* pathogenicity in cell cultures [7,67,68,69]. 

Most importantly, the bacterial surface LOS has been identified as an essential virulence factor of *C. jejuni* playing key roles in host cell adhesion and invasion as well as in the attraction and accumulation of macrophages and granulocytes to the sites of intestinal bacterial invasion [11,24,39,40,41]. In vitro studies revealed that sialylation of *C. jejuni* LOS enhances the invasive potential of the pathogen and may further contribute to the evasion of attacking antibodies [39,70,71]. Since the distinct *C. jejuni* LOS structure (Figure 2) is associated with campylobacteriosis severity and post-infectious sequelae such as GBS, a better understanding of the molecular mechanisms underlying both intestinal and extra-intestinal disease manifestations is of basic importance for the development of novel strategies in order to prevent or treat human *C. jejuni*-induced disease. 

## 4. *C. jejuni* Lipooligosaccharide and Post-Infectious Sequelae in Human Campylobacteriosis

The results from several clinical studies demonstrate that the hyper-activation of the innate immune system by sialylated *C. jejuni* LOS triggers the development of severe forms of campylobacteriosis and increases the risk for development of post-infectious sequelae such as GBS [22,72] (Figure 2) The majority of *C. jejuni* strains produce LOS mimicking distinct surface structures of human gangliosides [73]. Antibodies directed against LOS hence might react with human peripheral nerves following *C. jejuni* infection [74]. Interestingly, GBS itself is triggered by these autoantibodies produced in response to LOS of *C. jejuni* strains [75]. Specifically, *C. jejuni* strains of the classes A, B and C, known as the ABC group, received lots of attention due to their possession of coding genes for the sialic acid-processing enzymes and glycosyltransferases, which are required for the synthesis of ganglioside-like glycans [45,71,76]. In one study, the LOS outer-cores of 26 *C. jejuni* strains (22 associated with GBS, four associated with MFS) were analyzed via mass spectrometry and revealed that almost all strains expressed LOS with structural similarities to gangliosides in nerve fibers [73]. Genes for production of sialylated LOS classes have been found predominantly in GBS-associated *C. jejuni* strains [77]. Furthermore, it has been suggested that *C. jejuni* strains with sialylated LOS have a higher invasive potential in cell cultures, and that inactivation of LOS sialyltransferase results in the loss of such invasiveness [78]. However, a recent study reported that the sialylation of LOS is not necessary for the *C. jejuni* invasiveness, given that only 23% of isolates from patients with *C. jejuni* bacteremia belonged to LOS locus classes indicative of a potential to sialylate the LOS [55]. 

Together, these findings support the view that *C. jejuni* LOS constitutes a major pro-inflammatory trigger for severe forms of human campylobacteriosis and plays an important role in the immunopathogenesis of post-infectious sequelae such as GBS and MFS [45]. 

## 5. Structural Aspects of *C. jejuni* Lipooligosaccharide 

The *C. jejuni* LOS consists of the hydrophobic lipid A anchor and an oligosaccharide with a conserved inner and a variable outer core (Figure 2) [39,79]. The LOS biosynthesis genes in *C. jejuni* are localized at a hypervariable locus with 19 identified classes [80], which may explain the variation in *C. jejuni*-associated pathologies [81,82]. 

Molecular analyses demonstrated that the *C. jejuni* GB11 strain (isolated from a GBS patient) is genetically similar to the completely sequenced *C. jejuni* NCTC 11168 strain, whereas the expressions of LOS genes were shown to strongly vary between respective strains. The presence of the identical LOS locus in both *C. jejuni* ATCC 43446 and GB11 strains is probably the result of horizontal gene transfer [72]. Additionally, the serum from the GB11 strain infected patient reacted only with the LOS of GB11 and ATCC 43446 strains, but not with the LOS of the NCTC 11168 strain [72]. These findings indicate that the horizontal exchange of genetic material between *C. jejuni* strains leads to variations in LOS synthesis genes, which might be indeed related to *C. jejuni*-associated immunopathologies.

Interestingly, surface carbohydrate structures such as LOS play essential roles as permeability barriers against hydrophobic antibiotics [83] and phenolic antimicrobials [84]. The inactivation of the *C. jejuni waaF* gene (encoding a bacterial heptosyltransferase) results in the truncation of the LOS inner core [85], causing a reduction in the minimal inhibitory concentrations (MICs) of phenolic antimicrobials, such as butylated hydroxytoluene, *p*-coumaric acid, epigallocatechin gallate, and hesperidin [84]. Furthermore, results from another study with mice that had been infected with *C. jejuni* Δ*lgtF*, Δ*waaF,* and parental strains revealed that LOS may be involved in bacterial colonization properties [24]. However, cultivation of the *C. jejuni* mutant strains showed that the growth of both mutants was unaffected, indicating that the observed colonization defects of the Δ*lgtF* and Δ*waaF* mutant strains might indeed constitute specific and interesting in vivo phenomena [24].

As mentioned above, LOS is essential for both cellular invasion and translocation of *C. jejuni* across intestinal epithelial barriers [11,24,39,40,41,78]. The inactivation of sialyltransferase in GBS-associated *C. jejuni* strains (expressing sialylated LOS) resulted in less distinct invasiveness into human epithelial cells as compared to the respective wildtype bacteria [39]. In addition, *C. jejuni* isolates expressing LOS mimicking ganglioside structure were shown to strongly attach to the Caco-2 intestinal epithelial cells, as opposed to *C. jejuni* control isolates [78]. This is of clinical relevance given that *C. jejuni* ganglioside-like LOS-expressing isolates are linked to severe enterocolitis with bloody diarrhea [78].

Finally, the variation of the LOS structure that may even mimic human ganglioside-like structures by *C. jejuni* is a strategy to evade recognition by the host immune system. This mechanism is common to bacterial pathogens in general.

## 6. Activation of TLR-4 Signaling by *C. jejuni* Lipooligosaccharide in Humans and Mice

Similar to LPS from different bacteria vertebrate species such as humans and mice recognize *C. jejuni* LOS by specific binding of the lipid A moiety to the TLR-4 receptor complex of the innate immune system (Figure 3). The TLR-4 signaling pathways have been extensively reviewed by others [86,87,88]. Remarkably, mice have been shown to be approximately 10,000-fold more resistant to LOS as compared to humans. Briefly, TLR-4 induces intracellular signaling through at least two major pathways: (I) the Toll-Interleukin Receptor (TIR)-domain-containing interferon-β (TRIF), TRIF-related adapter molecule (TRAM), TRIF–TRAM-pathway, which upregulates genes encoding type I interferons (IFNs), and activates tumor necrosis factor (TNF)-α production and secretion; and (II) the TIR-domain containing adapter protein (TIRAP), Myeloid differentiation primary response 88 (MyD88), TIRAP–MyD88-pathway, which regulates early nuclear factor kappa-light-chain-enhancer of activated B cells (NF-κB) activation and related inflammatory cytokine production such as interleukin (IL)-12, responsible for the majority of the LPS/LOS responses [12,88,89,90] (Figure 3). 

The immune responses induced by LPS or LOS variants of different Gram-negative bacteria upon TLR-4 activation have been shown to differ remarkably [91]. While the LPS of the *Helicobacter pylori* bacteria that are closely related to *C. jejuni* shows low reactivity with TLR-4 [92], the lipid A moiety of *C. jejuni* LOS acts as a highly potent TLR-4 agonist [93]. Additionally, differences in the expression and activation of TLR-4 by bacterial LPS or LOS were observed between vertebrate species (e.g., rabbits, swains, dogs, humans, and mice). In mice, as in humans, cells of myeloid origin (such as monocytes, macrophages and granulocytes) exhibit higher levels of TLR-4 expression than lymphoid cell types, such as T and B cells [88]. However, murine plasmacytoid dendritic cells (pDCs) express TLR-4 in contrast to human pDCs [94].

The main difference between murine and human TLR-4 activation is most probably due to the LPS/LOS recognition pattern of the receptor complex [88]. Human, as opposed to murine, TLR-4 can differentiate between the hexa- and penta-acylated forms of LPS produced by *Pseudomonas aeruginosa*, for instance [95]. Furthermore, mice and humans differ regarding their TLR-4-expression levels upon LPS challenge. While LPS increases the TLR-4 expression in human macrophages and monocytes [96], murine peritoneal macrophages and neutrophils have shown decreased TLR-4 expression levels [97]. Additionally, the LPS/LOS doses being used in mice (1–25 mg/kg) are about 100-10,000 times higher than the concentrations that are needed to induce severe disease including shock in humans [44]. These observations underline the fact that the LPS/LOS tolerance of rodents, such as mice and rats, renders those animals highly resistant to bacterial LPS and LOS [44,97]. 

Interestingly, there are several natural plant products such as curcumin derived from *Curcuma longa* acting as highly effective TLR-4 antagonists by non-activate binding leading to competitive inhibition of LPS/LOS functions [98]. Recently, it has been demonstrated that apoptosis induction, tight junction redistribution, and increased inflammatory responses upon *C. jejuni* infection were down-regulated after curcumin treatment in a co-culture model of human colon epithelial cells HT-29/B6-GR/MR and immune THP-1 cells [99]. Additionally, curcumin could effectively ameliorate the campylobacteriosis symptoms including apoptosis, T-cell mediated inflammatory responses and colonic barrier dysfunction, observed in *C. jejuni* infected IL10^−/−^ mice [99]. Furthermore, the increased concentrations of pro-inflammatory cytokines such as TNF-α, IL.-1β and IL-6 in *C. jejuni* infected co-cultures were down-regulated after curcumin treatment [99]. These results indicate that curcumin in competition to TLR-4 binding antagonized local and systemic inflammatory responses triggered by bacterial LPS/LOS [99,100]. Moreover, it has been reported that the competition between curcumin and LPS/LOS to bind TLR-4 inhibited the MyD88-dependent pathway [101]. These results provide strong evidence that the activation of TLR-4, and, thus, inflammatory diarrhea due to *C. jejuni* LOS induced immune responses may be dampened by a curcumin enriched diet. 

## 7. Generation of Secondary Abiotic Mice to Overcome Colonization Resistance

Several animal models, such as *Galleria mellonella larvae* [102], colostrum-deprived piglets [103], gnotobiotic piglet [104], naïve swines [105], and ferrets [106] were successfully used for the study of numerous aspects of *C. jejuni* pathogenesis. However, each animal model showed strong limitations in terms of employing or comparing with human conditions upon *C. jejuni* infection [54]. The use of mice as a model for *C. jejuni* infection was banned in the early 2000s, given that conventional wildtype mice are highly resistant to *C. jejuni* colonization due to the physiological colonization resistance (CR) exerted by the murine gut microbiota composition [43,46,50]. 

On the other hand, it has been known for decades that *C. jejuni* is able to colonize the intestines of germfree BALB/c and C57BL/6 mice [56,59,60]. Table 1 (conventional and microbiota depleted wild-type mice models) and 2 (genetically modified mice models) summarize various murine in vivo studies, which are further described in detail in the text. In support, our studies revealed that *C. jejuni* are able to colonize the gastrointestinal tract of secondary abiotic wildtype mice and secondary abiotic mice that had been associated with a complex human gut microbiota following FMT, resulting in pro-inflammatory immune responses in the colon upon peroral infection [43,46,107]. Further investigations demonstrated that murine CR against *C. jejuni* is abrogated by modifying the microbiota composition towards increased intestinal commensal *Escherichia coli* loads, as seen following *E. coli* feeding via the drinking water, during intestinal inflammation or in infant mice immediately after weaning [58,108]. In the neonatal mouse model, *C. jejuni* was even able to colonize and cause clinical signs such as severe diarrhea with increased mucus discharge, reduced weight gain, and sometimes bloody stool, as early as 5 days post-infection [57], which is in line with a study by Lotz et al., demonstrating distinct differences between fetal, neonatal, and adult murine intestinal epithelial cells, particularly regarding TLR-4 expressing features [109]. These findings revealed that CR and the resilience to *C. jejuni* LOS are not developed in the early stage of murine life, as is discussed further below (Chapter 9).

## 8. Rendering Mice Susceptible to LOS as Approach for Developing Valid *C. jejuni* Infection Models 

Stahl et al. developed an in vivo *C. jejuni* infection model through application of a single antibiotic compound to mice [54]. Peroral application of vancomycin was accompanied by a depletion of *Bacteroidetes* and *Clostridia,* and conversely, a promoted growth of *Lactobacilli* within the murine intestinal lumen [54,110]. These alterations of the murine microbiota composition allowed *C. jejuni* to colonize the intestinal lumen at high loads. Based on the finding that upon *C. jejuni* infection vancomycin treated mice that were deficient in Single Ig IL-1-related receptor (SIGIRR) develop intestinal inflammation, the authors proposed *Sigirr*^−/−^ mice as a novel campylobacteriosis model. Following peroral *C. jejuni* infection, *Sigirr*^−/^^−^ mice exerted more severe gross pathology as compared to wildtype mice, which was accompanied by higher expression levels of pro-inflammatory cytokines such as TNF-α, IFN-γ, and IL-17 in the former versus the latter [54]. Furthermore, in this model, even high load pathogenic colonization of the intestinal tract was accompanied by a relatively moderate and self-limiting course of *C. jejuni* induced inflammatory disease [54,111]. In line with the high LOS tolerance of mice discussed above, activation of murine as well as human SIGIRR inhibits the MyD88-dependent signaling pathways [112] and, thus, the SIGIRR deficiency sensitizes mice to *C. jejuni* LOS. 

Similar to SIGIRR deficiency, the lack of the anti-inflammatory cytokine IL-10, which inhibits TLR-dependent innate immune responses, abolished the murine resistance to LOS and LPS [44,50,113,114,115,116]. It has recently been shown that *C. jejuni* colonize the intestinal tract of five to six months old IL-10^−/−^ mice with murine intestinal microbiota suffering from chronic colitis; however, the clinical signs for severe human campylobacteriosis such as bloody diarrhea were missing [50,61]. Inspired by the initial use of IL10^−/−^ mice as a clinical model for campylobacteriosis developed by Mansfield et al. [56] and Lippert et al. [117], our group has generated a similar experimental model for severe campylobacteriosis. For this purpose, IL10^−/−^ mice were subjected to broad-spectrum antibiotic treatment starting immediately after weaning in order to deplete the commensal gut microbiota and, hence, to abrogate physiological CR preventing *C. jejuni* infection and furthermore, in order to eliminate potential colitogenic stimuli from the commensal gut microbiota, leading to chronic colitis in IL10^−/−^ mice after 3 months or more [47,48,50]. Within one week following peroral *C. jejuni* infection, these so-called secondary abiotic IL10^−/−^ mice were stably colonized with high pathogenic loads and suffered from non-self-limiting acute enterocolitis, characterized by wasting and bloody inflammatory diarrhea, thus mimicking key features of severe human campylobacteriosis [50]. In addition, both *C. jejuni*-induced innate and adaptive pro-inflammatory immune responses were not limited to the intestinal tract but could also be observed in extra-intestinal including systemic compartments. The isolation of *C. jejuni* from mesenteric lymph nodes (MLNs), liver, kidneys, spleen and cardiac blood indicated that invading bacteria had been able to translocate from the inflamed intestines to extra-intestinal and even systemic compartments, further perpetuating the fatal inflammatory scenario. Overall, the secondary abiotic IL-10^−/−^ mouse model has been proven valuable to unravelling the immunopathological impact of distinct *C. jejuni* virulence factor involved in human disease [28,43,47,48,52,64]. 

Remarkably, TLR-4 and IL10 double deficient (TLR-4^−/−^ IL-10^−/−^) mice were far less compromised and exhibited less distinct intestinal as well as extra-intestinal including systemic sequelae upon *C. jejuni* infection. These results provide further evidence for a major role of TLR-4 mediated, LOS dependent immunopathological mechanisms underlying campylobacteriosis [46,50]. In support, *C. jejuni* infection experiments in TLR-4 deficient *Sigirr*^−/−^ mice revealed similar results [54]. Following peroral infection of TLR4^−/−^
*Sigirr*^−/−^ mice, *C. jejuni* colonized the intestines at high levels, but induced rather mild, if any, signs of enteritis. Compared to *Sigirr*^−/−^ counterparts, TLR-4 deficient *Sigirr*^−/−^ animals displayed less TNF-α and IFN-γ expression in ceca upon infection, which was comparable to that assessed in uninfected control mice [54]. 

Recently, Giallourou et al. reported that zinc deficiency had a direct impact on symptom formation in response to *C. jejuni* infection in antibiotic pretreated mice. In this mouse model, *C. jejuni* infection induced both, intestinal and systemic inflammatory responses, underlining the relevance of this model in clinical campylobacteriosis research [53]. In line with the fact that zinc deficiency increases the sensitivity of mice to *C. jejuni* LOS, it is of note that significantly decreased serum zinc concentrations were measured in patients with various bacterial infections [118,119], indicating a negative correlation between zinc and susceptibility to infection-related disease (Figure 4). In support of this, challenging rats with LPS of *E. coli* resulted in decreased serum zinc concentrations [120]. However, intraperitoneally injected zinc protected mice against lethality that was induced by *Salmonella typhimurium*-derived LPS injection [121]. Due to the anti-apoptotic [122] and anti-inflammatory properties of zinc [123,124,125], its supplementation during pregnancy protected mice from *E. coli*-LPS induced preterm delivery and fetal death by inhibiting LPS induced TNF-α release as well as NF-κB and mitogen-activated protein kinase activation [126]. In support, polaprezine, a chelating compound of zinc and L-carnosine, inhibited endotoxin shock in mice that had been challenged with LPS through inactivation of NF-κB and, subsequently, reduced nitric oxide and TNF-α releases [127]. Additionally, zinc deficiency was associated with increased LPS-induced TNF-α and IL-10 concentrations in mesenteric leukocytes isolated from rats with dextran sulfate sodium induced colitis [128]. Notably, decreased plasma levels of zinc could be measured in patients suffering from Crohn’s disease [129]. The molecular detection of *C. jejuni* via real-time polymerase chain reaction in gastrointestinal biopsies revealed that patients suffering from Crohn’s disease were more prone to *C. jejuni* infections as compared to individuals without preexisting intestinal morbidities [130], which might be related to decreased zinc levels in these patients [118]. Hence, zinc may be considered as a pharmacological agent particularly against Gram-negative bacterial infection and to reduce infection burden via its immune-modulatory (i.e., anti-inflammatory) properties. 

In summary, the abolishment of CR against *C. jejuni* in mice by depletion or modification of the host specific gut microbiota in line with the sensitization of the animals to *C. jejuni* LOS by genetic manipulation and zinc deficiency was crucial for the development of appropriately modified murine infection models mimicking key features of campylobacteriosis and, thus, providing a better understanding of cellular and molecular mechanisms underlying *C. jejuni* induced human disease (Figure 4). Additionally, these results prove that TLR-4 dependent LOS signaling is essential for *C. jejuni*-induced intestinal immunopathology in mice (Figure 3) and that innate immune responses represent hallmarks in the immunopathogenesis of campylobacteriosis [46,50] (Figure 1). The extreme resistance of birds to LPS/LOS might help to explain the lack of *C. jejuni*-mediated diseases in domestic chickens, as is discussed in more detail below (Chapter 10). 

## 9. The Use of Novel Murine Models of *C. jejuni* Infection in Actual Campylobacteriosis Research

Many recent studies investigated various immunopathological aspects of campylobacteriosis, including *C. jejuni*-induced host responses and virulence factors in genetically modified mice with or without a specifically modified gut microbiota. As mentioned previously, it has been shown that the murine microbiota is an essential physiological prerequisite for an effective CR directed against *C. jejuni*. Besides secondary abiotic mice, mice harboring a human gut microbiota, infant mice and mice displaying elevated commensal intestinal *E. coli* loads constitute valuable tools for investigating *C. jejuni*-host interactions and infection-induced immunopathological responses of varying severities [47,48,131].

Given the central role of IL-10 in maintaining gut homeostasis by the suppression of inflammatory responses [132,133], it is also of great importance that conventional infant mice, which did not raise a sufficient IL10 response in their intestines, become effectively colonized and infected by *C. jejuni*. Most importantly, upon *C. jejuni* infection those infant mice display the typical course of self-limiting campylobacteriosis seen in humans [56]. Moreover, infant mice cleared the intestinal infection and developed granulomas at extra-intestinal sites including liver, kidneys and even the lungs. These features highlight conventional infant mice as a useful model to study campylobacteriosis and possibly the autoimmune responses leading to post-infectious sequelae. 

In particular, the LOS sensitized secondary abiotic IL-10 deficient mice were successfully used for groundbreaking studies for a better molecular understanding of immunopathogenesis of campylobacteriosis. In this model, murine symptoms mimic human disease and induction of intestinal inflammation strictly depends on the motility and invasive properties of *C. jejuni* [28]. It is of note that commensal *E. coli* lacking any invasive or other pathogenic properties do not induce pathology in this model [50]. This murine model was used recently for the successful treatment of *C. jejuni* infection by the potent immune suppressor sirolimus (rapamycin), which inhibits intracellular signaling by mammalian target of rapamycin (mTOR). The mTOR complexes 1 and mTORC2 play important roles in regulating the activation of innate immune cell populations such as macrophages and dendritic cells and induce cytokine production [134,135]. Therefore, mTOR signaling affects several innate signaling pathways involved in TLR-4 activation and IL-10 production [135,136,137]. Additionally, mTOR-signaling is crucial for cell growth, proliferation, bacterial killing, and clearance of infections [136]. In addition to mTOR, the phosphatidylinositol 3-kinase-γ (PI3K-γ, upstream of mTOR) have been shown to modulate *C. jejuni*-induced colitis independently of T cell activation by recruitment of neutrophilic granulocytes to the infected intestines and subsequent massive TNF secretion [62,138]. By using germ-free IL-10 and Rag2 double deficient mice the authors confirmed the hypothesis that innate immune cells are the main cellular compartment responsible for campylobacteriosis. Most importantly, inhibition of PI3K-γ signaling ameliorated *C. jejuni*-induced neutrophil accumulation which correlated with reduced inflammatory responses. In fact, neutrophils have been shown to exert potent pro-inflammatory responses upon LPS stimulation in vitro characterized by TLR-4 dependent TNF-α and IL-6 secretion [63,138]. To determine the role of mTOR in modulating TLR-4 activation, bone marrow-derived neutrophils were cultured with the mTOR inhibitor rapamycin [63]. Upon LPS stimulation, a reduction in TNF-α and IL-6 production by neutrophils could be observed, further underlining the regulatory role of mTOR within the TLR-4 dependent LPS/LOS signaling pathway [63]. Moreover, under the control of mTOR signaling, *C. jejuni* has been shown to produce a cytolethal distending toxin, a genotoxin with DNase activity that is supposed to be involved in the pathogenesis of colorectal cancer [139,140,141]. The inhibition of mTOR and, thus, the production of genotoxin through rapamycin, could sufficiently dampen *C. jejuni*-induced carcinogenesis [139]. Hence, these results suggest that *C. jejuni*-induced colitis or colorectal tumorigenesis can be attenuated by inactivation of mTOR signaling using pharmacological interventions such as rapamycin or PI3K-γ inhibitors [62,139]. The mTOR signaling pathways represent a valuable novel target for therapeutically interventions and drug development, enabling modulation of innate immune responses for treatment of campylobacteriosis. 

In the past few years, secondary abiotic as well as conventional NOD2 deficient IL-10^−/−^ (NOD2^−/−^ IL-10^−/−^) mice were used to unravel the role of the innate immune receptor nucleotide-oligonucleotide-domain 2 (NOD2) during campylobacteriosis and clearance of *C. jejuni* infection [51,142,143,144,145,146]. Following *C. jejuni* infection, secondary abiotic NOD2^−/−^ IL-10^−/−^ mice harbored the pathogen at high loads, but developed less severe enterocolitis as compared to IL-10^−/−^ counterparts [51], indicating that NOD2 signaling increases the severity of campylobacteriosis. Conflicting data from another study revealed that NOD2 signaling inhibits murine campylobacteriosis [142]. These inconsistent results might be explained by differences in the genetic background of mice as well as in the experimental set-ups between studies including the presence or absence and distinct composition of the intestinal microbiota (e.g., antibiotic treatment duration, different methods and the time point of sampling post-infection). The important role of the microbiota composition in campylobacteriosis including the induction of neurological sequelae in frame of GBS has recently been confirmed in elegant studies which paved the way for the use of murine models for the study of immunopathology of *Campylobacter* induced GBS [147,148,149].

Even though the exact roles of NOD2 during campylobacteriosis remain unknown, several reports have emphasized the complex modulatory crosstalk between NOD2 and TLR-4 signaling especially in pro-inflammatory cytokine regulation [150,151,152,153,154], given that NOD2 is able to regulate the IL-12 production depending on TLR-4 intensity [155]. Furthermore, a clinical study revealed that a distinct NOD2 gene mutation (homozygous for the 3020insC mutation) observed in Crohn’s disease patients leads to massive TNF-α release by TLR-4 activation via LOS or LPS [156], indicating that a mutation in the NOD2 gene is associated with increased LOS susceptibility. In line, NOD2 is able to inhibit TLR-2 and -4 mediated induction of pro-inflammatory cytokine secretion [157,158]. Taken together, these results confirm the down-regulation of TLR-4 signaling by NOD2 activation not only in IBD [159], but also in campylobacteriosis.

Most recently, the secondary abiotic IL10 deficient murine model of campylobacteriosis was standardized and could be developed to the preclinical level for pharmaceutical analysis of alternative drugs, including ascorbate and vitamin D, which effectively suppressed *C. jejuni*-induced inflammation in the course of campylobacteriosis [20,160].

## 10. Differences in LOS Tolerance among Humans and Birds Impede the Development of Prophylactic Measures against Campylobacteriosis

The major role of the endotoxin LOS in the inflammatory response to *C. jejuni* in both humans and mice hampers the development of vaccines against *C. jejuni*. This is mainly due to the fact that it is much more difficult to develop effective vaccines against endotoxin-driven morbidities as compared to exotoxin-mediated diseases [74]. In addition, as mentioned previously, due to structural similarities between *C. jejuni*-LOS and human gangliosides, antibodies directed against LOS might react with human peripheral nerves [73,74]. Therefore, the association of anti-*C. jejuni* LOS antibodies with GBS constitutes a substantial challenge if one might consider LOS a potential target molecule for vaccine development. Likewise, the production of antibodies directed against *C. jejuni* flagella could be difficult since the corresponding structural proteins are heavily glycosylated and show massive structural variations among bacterial strains [161]. However, the capsular polysaccharide (CPS) might be regarded as a promising candidate for conjugate vaccine development against *C. jejuni*. The advantage, as opposed to LOS, is that neither mimicry between gangliosidic surface structures and *C. jejuni* CPS nor association of CPS with GBS have been described to date [162]. Additionally, a mutated CPS (i.e., *kpsS* mutant) has been shown to increase the susceptibility of *C. jejuni* to antibiotic and phenolic compounds with antibacterial properties [84]. The *kpsS* gene encodes a β-linked 3-deoxy-d-*manno*-oct-2-ulosonic acid (Kdo) transferase playing an important role in CPS synthesis [163]. Moreover, in the antibiotic pretreated zinc deficient mouse model (described by Giallourou et al.), a *C. jejuni kps* mutant strain was unable to colonize the murine intestines as opposed to the wildtype counterpart [53]. Additionally, a monovalent capsule conjugate vaccine against *C. jejuni* has provided 100% protection against diarrheal disease in monkeys [164]. However, since 47 different CPS serotypes have been identified for *C. jejuni* so far [165], the development of capsule based vaccines needs further challenging investigations.

Human *Campylobacter* infections result in severe inflammation of the intestinal mucosa, whereas colonized avian species harbor *C. jejuni* as intestinal commensal bacteria and are usually lacking any clinical signs [12]. In order to decrease the *C. jejuni* infection rates in humans, and in consequence, the prevalence of severe human campylobacteriosis with the risk of subsequent post-infectious sequelae, the vaccination of broiler chickens against *C. jejuni* is proposed to be one of the most efficient strategies to reduce the intestinal *C. jejuni* loads in chickens and, thus, the risk of transfer to humans via consumption of contaminated meat. The comparison of the *C. jejuni* colonization properties in humans and chickens revealed host-specific differences in susceptibility to the bacteria. For example, the characterization of chicken TLRs showed major species-specific characteristics [12]. *C. jejuni* induces IFN-β, a key mediator of systemic inflammation, through TLR-4 activation in humans, but not in chicken [12]. Furthermore, *C. jejuni* is highly susceptible to the chicken host defense peptide cathelicidin-2 exerting antimicrobial activities [166]. These results may explain why *Campylobacter* is able to colonize the avian gastrointestinal tract (with up to 10^9^ CFU/g feces) without causing overt clinical signs [166,167,168]. Since the *C. jejuni*-induced immune responses in humans and chicken are very different, applying vaccines against *C. jejuni* to chickens could only reduce the *Campylobacter* transfer to humans. However, there are several studies using clinical strains such as *C. jejuni* 81-176 in chicken for vaccine development strategies [165,169]. 

Results from an in vivo study with Ross broiler chickens revealed four promising antigens (YP437, YP562, YP9817 and YP9838) as potential vaccine candidates against the *C. jejuni* 81-176 strain [169]. The identified proteins were either flagella-associated proteins or hypothetical proteins. The administration of these antigens could reduce cecal *C. jejuni* loads by up to 4.2 log orders of magnitude in one trial. However, these promising results could not be reproduced, despite a strong induced immune response upon application of the same antigen [169]. In addition, recent investigations suggest *C. jejuni* N-glycan as a potential antigenic structure for vaccine development, since it constitutes a surface carbohydrate with immunogenic properties in rabbits and humans [170,171]. In fact, a recent study revealed the protective effect of N-glycan based vaccines against the *C. jejuni* 81-176 strain in SPF Leghorn chickens [165]. 

Finally, it is noteworthy that chickens are more resistant to endotoxin derived from Gram-negative bacteria, such as *C. jejuni,* as compared to mice and humans [166,172]. In contrast to humans, however, *C. jejuni* mainly locate in the mucous layer of the crypts in chickens and do not directly adhere to the avian intestinal epithelial cells [173,174,175]. Furthermore, *C. jejuni* rarely invade the intestinal epithelium and if the internal organs are affected, no clinical signs of disease are observed in colonized chickens [173,175,176]. Interestingly, the intravenous injection of high doses of purified *E. coli*-LPS (i.e., 577 mg per kg body weight) into chickens induced only rather mild clinical signs such as diarrhea and gasping, and the chickens recovered within two days post application. However, a LPS dose of 45.5 mg per kg body weight was lethal to mice within four hours [172]. In humans, 4 ng per kg body weight of Gram-negative bacterial endotoxin (e.g., *C. jejuni*-LOS) have been shown to activate pro-inflammatory immune cytokine responses [177]. In a clinical study, the administration of 1 mg *Salmonella minnesota* endotoxin to humans induced an inflammatory response with septic shock-like symptoms that continued for several days [178]. 

These findings further underline that differences in LOS susceptibility and resistance between vertebrate species such as humans on one hand and chicken and mice on the other might provide explanations for the completely divergent species-specific reactions to *C. jejuni*.

## 11. Conclusions

The major role of innate immunity and *C. jejuni* LOS as a main pathogenicity factor in human campylobacteriosis and the extreme LOS resistance of mice and birds has provided a substantial key to the understanding of symptomless *C. jejuni* colonization in chickens on one hand, versus severe infection in humans on the other. In line with a better appreciation of the CR created by the individual microbiota of vertebrates, these historically neglected but valid insights enabled the development of several novel murine models for campylobacteriosis research—all based on the reduction in species-specific resistance to both, bacterial colonization and LOS resistance. The resulting LOS sensitized murine *C. jejuni* infection models lacking CR mimic key features of human campylobacteriosis and will help to unravel the molecular mechanisms underlying the immunopathology of intestinal infection as well as post-infectious sequelae. Their standardization and use for preclinical pharmaceutical evaluation have paved the way for the identification of novel therapeutic and prophylactic options in order to combat campylobacteriosis, including collateral damages, in the future. 

## Figures and Tables

**Figure 1 microorganisms-08-00482-f001:**
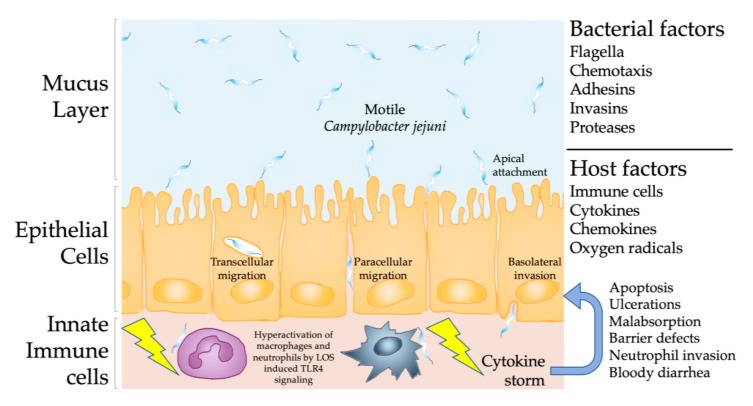
*Campylobacter jejuni* induced immunopathological responses within the intestinal tract. The highly motile *C. jejuni* circumvent the mucus layer, cross the intestinal epithelial layer and interact with mucosal and lamina propria cells, resulting in the recruitment of dendritic cells, macrophages, and neutrophils. The interaction of these innate immune cells with the lipooligosaccharide (LOS) of *C. jejuni* mounts a massive pro-inflammatory cytokine response via TLR-4 signaling, inducing apoptosis and ulcerations in the epithelial layer, progressing into barrier defects, malabsorption and bloody diarrhea.

**Figure 2 microorganisms-08-00482-f002:**
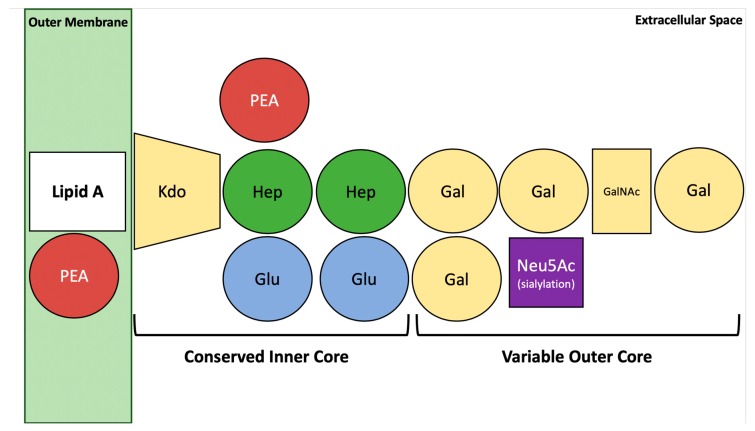
The basal structure of *C. jejuni* LOS. The LOS of *C. jejuni* consists of Lipid A, phostoethanolamine (PEA), 3-Deoxy-d-manno-oct-2-ulosonic acid or keto-deoxyoctulosonate (Kdo), Heptose (Hep), glucose (Glu), galactose (Gal), N-acetylgalactosamine (GalNAc), and N-acetylneuraminic acid (Neu5Ac) and in sialylated form mimics a range of gangliosides (e.g., GM1). The characteristic O-antigen of LPS from Gram-negative bacteria is missing.

**Figure 3 microorganisms-08-00482-f003:**
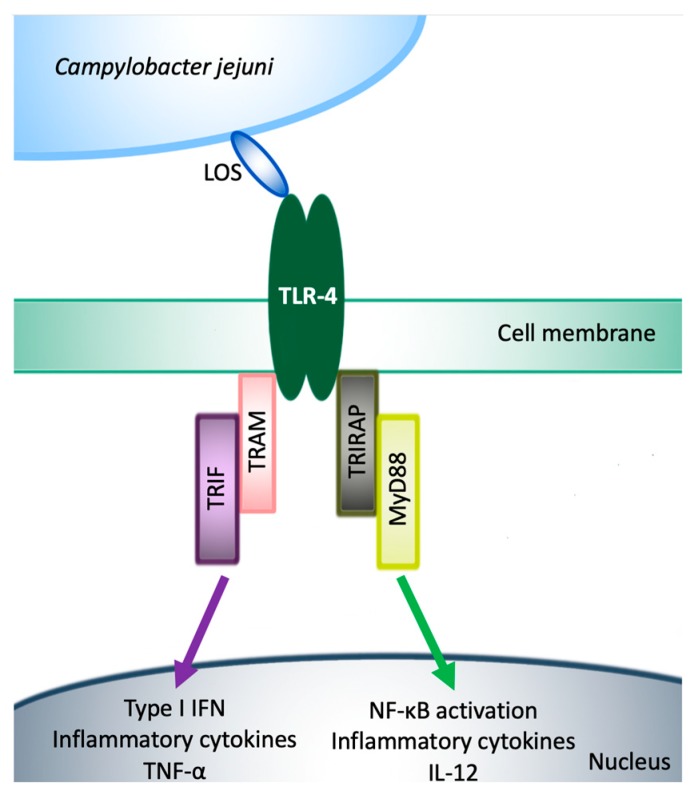
*C. jejuni* LOS and TLR-4 Signaling. The activation of TLR-4 leads to the synthesis of inflammatory mediators via two different pathways; TRIF-TRAM pathway, which induces the secretion of tumor necrosis factor alpha and TIRAP-MyD88, which is responsible for the LOS response and NF-κB activation.

**Figure 4 microorganisms-08-00482-f004:**
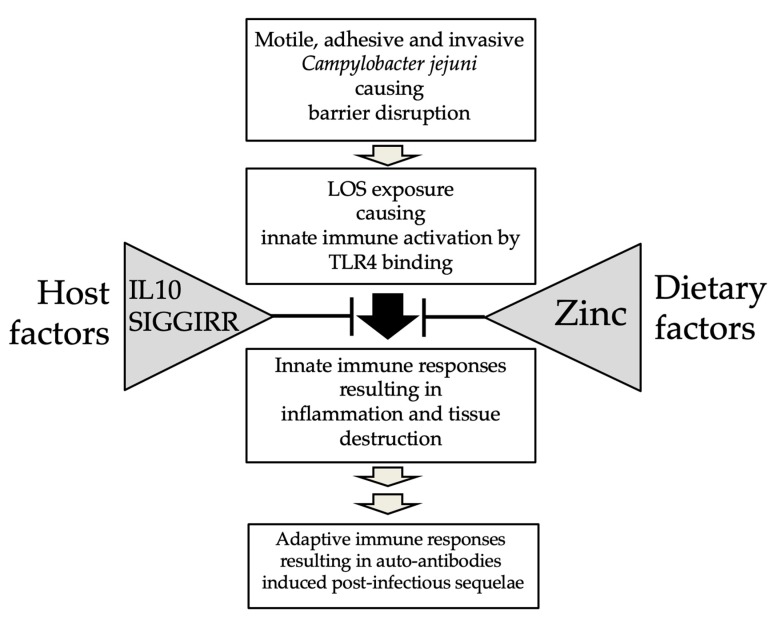
Modulation of *Campylobacter jejuni* infection by host and dietary factors. Both host and dietary factors interact with the immune responses upon *C. jejuni* infection at different levels thereby affecting the outcome of disease.

**Table 1 microorganisms-08-00482-t001:** Murine models of *C. jejuni* infection applying conventional and microbiota depleted wildtype mice.

Mouse Strain	Intestinal Microbiota	Antibiotic Treatment/Specific Diet	Intestinal Colonization of *C. jejuni*	LOS Sensitized	Macroscopic Signs of Disease	Inflammatory Responses	Reference
C57BL/6	SPF	No/No	Low	No	No	Elevated T-helper-1 cell responses in intestinal compartments	[56]
C57BL/6	SPF	No/No	No	No	No	Intestinal epithelial cell apoptosis and proliferation/regeneration, moderate cellular and molecular inflammatory responses	[46]
C57BL/6	No-SA	Yes/No	High	No	No	Intestinal epithelial cell apoptosis and proliferation/regeneration, pronounced cellular and molecular inflammatory responses	[46]
C57BL/6	SA plus rHF	Yes/No	Moderate	No	No	Intestinal epithelial cell apoptosis and proliferation/regeneration, pronounced cellular and molecular inflammatory responses	[46]
BALB/c Neonatal	SPF	No/No	Moderate	Yes	Wasting, bloody diarrhea	ND	[57]
C57BL/6 Infant	SPF	No/No	Moderate	Yes	Wasting, bloody diarrhea (self-limiting)	Intestinal epithelial cell apoptosis and proliferation/regeneration, elevated pro-inflammatory mediators, extra-intestinal T cell infiltrates	[58]
C57BL/6 Zinc deficient	Depleted	Yes/Zinc depleted	High	Yes	Wasting, bloody diarrhea	Intestinal edema, pronounced neutrophilic infiltration, crypt hyperplasia, intestinal and systemic inflammatory mediator responses	[53]

ND, not determined; SA, secondary abiotic; rHF, recolonized with human microbiota; SPF, specific pathogen-free murine microbiota.

**Table 2 microorganisms-08-00482-t002:** Murine models of campylobacteriosis applying genetically modified mice.

Strain/Species	Genetic Modification	Intestinal Microflora	Antibiotic Treatment	LOS Sensitized	Macroscopic Signs of Disease	Inflammatory Responses	Reference
BALB/c	Athymic (nu/nu) Euthymic (+/nu)	No, GF	No	Yes	Transient diarrhea	Cecal shrinkage, accumulation of eosinophils in the lower intestinal mucosa and lamina propria, translocation of viable *C. jejuni* to MLN, spleen, liver and kidneys	[59,60]
C57BL/6	IL-10^−/−^	SPF	No	Yes	Wasting, bloody diarrhea	Severe typhlocolitis and gross pathological changes in the gastrointestinal tract, increased immune cell responses and production of pro-inflammatory mediators	[56]
129/SvEv	IL-10^−/−^	No, GF	No	Yes	Wasting, bloody diarrhea	Intestinal crypt abscesses and epithelial ulcerations. Increased mononuclear cells, neutrophils, and elevated pro-inflammatory mediators	[61]
129/SvEv C57BL/6	IL-10^−/−^	No, GF	No	Yes	Wasting, bloody diarrhea	Pronounced neutrophilic infiltration of the intestines; increased NF-κB activity and Il1β, Cxcl2, Il17a expression	[62,63]
C57BL/10	IL-10^−/−^	No, SA	Yes	Yes	Wasting, bloody diarrhea	Increases in apoptotic intestinal epithelial cells. Pronounced innate and adaptive immune cell responses, elevated intestinal pro-inflammatory mediators. Translocation of viable *C. jejuni* to MLN, extra-intestinal including systemic inflammation	[50,64]
C57BL/6	SIGIRR^−/−^	Modified	Yes	Yes	ND	Increases in intestinal granulocytes, goblet cell depletion, edema, elevated pro-inflammatory mediators, translocation of *C. jejuni* to subepithelial tissues	[54]

ND, not determined; SA, secondary abiotic; SPF, specific-pathogen-free murine microbiota; GF, germfree; MLN, mesenteric lymph nodes.

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
