# Peer review of "Novel Clinical Campylobacter jejuni Infection Models Based on Sensitization of Mice to Lipooligosaccharide, a Major Bacterial Factor Triggering Innate Immune Responses in Human Campylobacteriosis"

_microorganisms, 2020, doi:10.3390/microorganisms8040482_

Round 1

Reviewer 1 Report

Review focuses on the role of C. jeuni lipooligosacharide (LOS) in campylobacteriosis and novel mouse models to study C. jejuni-induced disease.   Overall, this is a useful review which incorporates recent literature on this topic.  The major problem of the review is lack of any graphical information which makes it difficult for the reader to understand the concept. Inclusion of several figures and tables will greatly improve the clarity of the review.

Specific points:

1) LOS is one of major focus of the review. A figure describing how LOS of C. jejuni leads to invasion and subsequent disease will be very helpful. The figure can include structural LOS peculiarities, TLR2,4 > MyD88/TRIF signaling, immune activation> disease. It could be also two figures: one focused on structural aspects of LOS, and second - to signaling leading and disease pathogenesis.

2) Mouse models to study C. jejuni is another focus of the review. A table showing describes models to study C. jejuni infection in mice with relevant literature and advantages and limitation of each model will be very helpful.

Minor:

  1. Immune protective mechanisms to C. jejuni, including TLR and innate signaling could be better described in the review.
  2. There is no discussion of recent studies on type VI secretion system in C. jejuni pathogenesis.

Line 121-122. Sentence need to be rephrased.

Ref 9. DOI seems invalid.

Author Response

Reviewer 1:

Comments and Suggestions for Authors

Review focuses on the role of C. jeuni lipooligosacharide (LOS) in campylobacteriosis and novel mouse models to study C. jejuni-induced disease.   Overall, this is a useful review, which incorporates recent literature on this topic.  The major problem of the review is lack of any graphical information, which makes it difficult for the reader to understand the concept. Inclusion of several figures and tables will greatly improve the clarity of the review.

Specific points:

LOS is one of major focus of the review. A figure describing how LOS of C. jejuni leads to invasion and subsequent disease will be very helpful. The figure can include structural LOS peculiarities, TLR2,4 > MyD88/TRIF signaling, immune activation> disease. It could be also two figures: one focused on structural aspects of LOS, and second - to signaling leading and disease pathogenesis.

REPLY:

We thank the reviewer for the valuable suggestions. Respective information is now provided in more detail in Figure 1 and the new Figures 2 and 3.

 Mouse models to study C. jejuni is another focus of the review. A table showing describes models to study C. jejuni infection in mice with relevant literature and advantages and limitation of each model will be very helpful.

REPLY:

As suggested, we have added the new Table 1 summarizing the relevant murine models.

Minor:

Immune protective mechanisms to C. jejuni, including TLR and innate signaling could be better described in the review.

REPLY:

The following statement has been added (lines 233ff):

“Finally, the variation of the LOS structure that may even mimic human ganglioside-like structures by C. jejuni is a strategy to evade recognition by the host immune system. This mechanism is common to bacterial pathogens in general.”

There is no discussion of recent studies on type VI secretion system in C. jejuni pathogenesis.

REPLY:

A respective paragraph has been added (lines 71ff).

Line 121-122. Sentence need to be rephrased.

REPLY:

The sentence has been rephrased and now reads as (new lines 133ff):

“However, several studies suggest that the structure of C. jejuni LOS is indeed responsible for the outcome of infection and for the development of post-infectious sequelae [22,55].”

Ref 9. DOI seems invalid.

REPLY:

A valid DOI has been included.

Reviewer 2 Report

A thoroughly review on recent progress in the development of murine Campylobacter jejuni infection models. 

Some minor comments:

-Line 267: Delete the

-Lime 377 and 457: change "afore mentioned" to "mentioned previously"

-Avoid using "we" in the review paper. e.g: line 377=378 change to: "it has been shown that the murine microbiota..."

-Line 510-511: Move "however" to the front of the sentence

Author Response

Reviewer 2:

Comments and Suggestions for Authors

A thoroughly review on recent progress in the development of murine Campylobacter jejuni infection models. 

Some minor comments:

  • -Line 267: Delete the
  • -Line 377 and 457: change "afore mentioned" to "mentioned previously"
  • -Avoid using "we" in the review paper. e.g: line 377=378 change to: "it has been shown that the murine microbiota..."
  • -Line 510-511: Move "however" to the front of the sentence

REPLY:

We thank the reviewer for the constructive feedback.

All changes have been made as suggested.